# Longitudinal IgA and IgG Response, and ACE2 Binding Blockade, to Full-Length SARS-CoV-2 Spike Protein Variants in a Population of Black PLWH Vaccinated with ChAdOx1 nCoV-19

**DOI:** 10.3390/v15020448

**Published:** 2023-02-06

**Authors:** Muneerah Smith, Gaurav Kwatra, Alane Izu, Andrew Nel, Clare Cutland, Khatja Ahmed, Vicky Baillie, Shaun Barnabas, Qasim Bhorat, Carmen Briner, Erica Lazarus, Keertan Dheda, Lee Fairlie, Anthonet Koen, Shabir Madhi, Jonathan M. Blackburn

**Affiliations:** 1Department of Integrative Biomedical Sciences, Faculty of Health Sciences, University of Cape Town, Cape Town 7925, South Africa; 2South African Medical Research Council Vaccines and Infectious Diseases Analytics Research Unit (Wits-VIDA), Faculty of Health Sciences, University of the Witwatersrand, Johannesburg 2050, South Africa; 3Department of Clinical Microbiology, Christian Medical College, Vellore 632001, India; 4African Leadership in Vaccinology Expertise (ALIVE), Faculty of Health Sciences, University of the Witwatersrand, Johannesburg 2050, South Africa; 5Setshaba Research Centre, Tshwane 0182, South Africa; 6Family Centre for Research with Ubuntu, Department of Paediatrics, Stellenbosch University, Cape Town 7600, South Africa; 7Soweto Clinical Trials Centre, Soweto 6201, South Africa; 8Perinatal HIV Research Unit, Faculty of Health Science, University of the Witwatersrand, Johannesburg 2050, South Africa; 9Centre for Lung Infection and Immunity, Division of Pulmonology, Department of Medicine and UCT Lung Institute, University of Cape Town, Cape Town 7925, South Africa; 10Department of Infection Biology, Faculty of Infectious and Tropical Diseases, London School of Hygiene and Tropical Medicine, London WC1E 7HT, UK; 11Wits Reproductive Health and HIV Institute, Faculty of Health Sciences, University of the Witwatersrand, Johannesburg 2050, South Africa; 12Institute of Infectious Disease and Molecular Medicine, Faculty of Health Sciences, University of Cape Town, Cape Town 7925, South Africa; 13Sengenics Corporation, Level M, Plaza Zurich, Damansara Heights, Kuala Lumpur 50490, Malaysia

**Keywords:** COVID-19, SARS-CoV-2 spike protein, vaccine, HIV, IgA, IgG, neutralization

## Abstract

Vaccines against SARS-CoV-2 have been pivotal in overcoming the COVID-19 pandemic yet understanding the subsequent outcomes and immunological effects remain crucial, especially for at-risk groups e.g., people living with human immunodeficiency virus (HIV) (PLWH). In this study we report the longitudinal IgA and IgG antibody titers, as well as antibody-mediated angiotensin converting enzyme 2 (ACE2) binding blockade, against the SARS-CoV-2 spike (S) proteins after 1 and 2 doses of the ChAdOx1 nCoV-19 vaccine in a population of Black PLWH. Here, we report that PLWH (N = 103) did not produce an anti-S IgA response after infection or vaccination, however, anti-S IgG was detected in response to vaccination and infection, with the highest level detected for infected vaccinated participants. The anti-IgG and ACE2 blockade assays revealed that both vaccination and infection resulted in IgG production, however, only vaccination resulted in a moderate increase in ACE2 binding blockade to the ancestral S protein. Vaccination with a previous infection results in the greatest anti-S IgG and ACE2 blockade for the ancestral S protein. In conclusion, PLWH produce an anti-S IgG response to the ChAdOx1 nCoV-19 vaccine and/or infection, and ChAdOx1 nCoV-19 vaccination with a previous infection produced more neutralizing antibodies than vaccination alone.

## 1. Introduction

In response to the ongoing global 2019 coronavirus (COVID-19) pandemic, numerous vaccines have been developed with varying successes. The ChAdOx1 nCoV-19 vaccine [1,2,3,4] contains a replication-deficient chimpanzee adenoviral vector that contains the sequence information of the severe acute respiratory syndrome coronavirus 2 (SARS-CoV-2) structural surface glycoprotein. South Africa has a high human immunodeficiency virus (HIV) burden and recent findings suggests that COVID-19 increased the disease severity [5] for people living with HIV (PLWH). While Novavax NVX-CoV2327 vaccine efficacy was lower in PLWH (49%) compared to HIV-negative controls (60%) [6], safety and immunogenicity studies reveal that the ChAdOx1-nCoV19 vaccine was well tolerated in HIV-negative people and PLWH, and high responders retained neutralization to the B.1.351 (Beta; N501Y.v2) variant [4].

Since its emergence, the SARS-CoV-2 spike gene has accumulated numerous mutations that are major targets of the antibody response. Variants containing the N501Y mutation are associated with increased affinity to the angiotensin-converting enzyme 2 (ACE2) receptor, whereas variants containing the E484K and K417N mutations are associated with neutralizing antibody escape [7]. The B.1.1.7 (Alpha; N501Y.V1) variant was first identified in the United Kingdom and now also includes the E484K mutation. The B.1.351 variant was first identified in South Africa and contains 3 RBD mutations and 5 N-terminal mutations [8]. Pseudovirus neutralization assays indicate that 46% of convalescent serum samples were unable to neutralize the B.1.351 [9], which was corroborated in a live-virus neutralization assay neutralization assay [8].

In this study we employ a dual color microarray assay to assess the longitudinal Immunoglobulin A (IgA) and Immunoglobulin G (IgG) sero-response to the ChAdOx1 nCov-19 vaccine in a population of Black PLWH. We also employ microarray-based ACE2 blocking assay to determine how effective the vaccine is in producing antibodies that block ACE-binding to the ancestral, B.1.1.7 and B.1.351 S protein variants in Black PLWH.

## 2. Materials and Methods

### 2.1. Study Cohorts

This analysis forms part of randomized, double-blinded, placebo-controlled phase 1B/2A trial (COV005), which assesses the safety and immunogenicity of the ChAdOx1 nCoV-19 vaccine in South Africa [4]. Briefly, between 17 August 2020 and 12 November 2020, PLWH were enrolled for safety and immunogenicity monitoring, which was at the end of the first South African pandemic wave and the beginning of the second wave when the B.1.351 variant was detected in individuals (Appendix A). Blood samples were collected for serological analysis of SARS-CoV-2 infection from the longitudinal ChAdOx1 nCoV-19 vaccine trial of PLWH (N = 103), from seven South African research centers, hospitals and clinical trial centers in accordance with the principles of the Declaration of Helsinki and Good Clinical Practice Guidelines (Voysey et al., 2021), and stored at −80 °C until serological analysis. The time course includes the collection of bloods from day 0 (D0) i.e., before vaccination (N = 171); day 28 (D28) i.e., 28 days after the first dose of the vaccine or placebo (N = 170); and day 42 (D42) i.e., 14 days after the second dose of the vaccine or placebo (N = 166). The participants were further classified based on a previous SARS-CoV-2 infection, where a previous infection is defined as participants who were seropositive to the N protein at Day 0 according to the PPD study or had a reverse transcriptase polymerase chain reaction (RT-PCR) positive result prior to Day 0. Any participants who had a positive or negative RT-PCR result between D0 and D42 were excluded. The demographic data for the PLWH are in Appendix A. The participants were originally tested for SARS-CoV-2 using RT-PCR, using upper respiratory tract samples.

The COV005 study was approved by the South African Health Products Regulatory Authority (ref: 20200407) and the ethics committees of the University of Oxford (ref: OxTREC 35-20), University of the Witwatersrand (ref 200501), Stellenbosch University (ref M20/06/009_COVID-19), and University of Cape Town (ref: 350/2020). The COV005 study is registered with ClinicalTrials.gov, NCT04444674, and the Pan African Clinical Trials Registry, ACTR202006922165132. The trial protocol (version 6.0) is available online [4]. All patients provided written, informed consent.

For the serological assay set up, the pre-pandemic serum from patients who had active tuberculosis (TB) or active TB and HIV were utilized. The IgA and IgG serological assays included 46 pre-pandemic samples, and the ACE2 binding blockade assays included 27 pre-pandemic samples. The pre-pandemic bloods were collected in Cape Town, South Africa from 2011 to 2012, from a study approved by the HREC of the University of Cape Town (158/2010).

### 2.2. Printing the SARS-CoV-2 Spike Variant Protein Microarrays

Printing details are available in the Appendix A. Briefly, the SARS-CoV-2 S variant (ancestral, B.1.1.7, and B.1.351) protein microarrays were printed in a 24-plex format on streptavidin-coated glass slides (Nexterion, Schott) in a 3 × 3 microarray pattern. The slides were subsequently blocked and stored at −20 °C until it was used in assays.

### 2.3. Microarray Assays: Detecting IgA and IgG in Serum Samples

IMMUsafe slides (Sengenics Corporation, Singapore) were removed from −20 °C storage and blocked in blocking buffer (25 mM HEPES (MERCK), 50 mM KCl (MERCK), 20% glycerol (MERCK), 0.1% Triton X-100 (MERCK), 1 mM DTT (MERCK) and 50 µM biotin (Glentham Life Sciences, Corsham, UK) for 1 h. The slides were washed twice for 5 min in wash buffer (PBS and 0.2% Tween-20) and twice for 5 min wash in PBS (Gibco) only, then dried by centrifugation (Megafuge 40R, Thermo Scientific, Waltham, MA, USA) at 1200× *g* for 2 min, and assembled in cleaned 24-plex gaskets (GraceBio). The serum samples were thawed on ice and diluted 1:50 in incubation buffer (PBS, 0.2% Tween-20, 1% BSA and 1% milk powder (MERCK), and then incubated on the microarray for 1 h at room temperature shaking at 100 rotations per minute (RPM) (Orbital Shaker SO3, Stuart Scientific). The slides were subsequently washed thrice for 5 min in wash buffer, and incubated with 1.25 µg/mL Alexa fluor 555-labelled anti-human IgA and 1.25 µg/mL Alexa fluor 647-labelled anti-human IgG (ThermoFisher) (in incubation buffer) for 30 min. Microarrays were rinsed thrice, and then twice for 5 min in Wash buffer and twice for 5 min wash in PBS only, then dried by centrifugation at 1200× *g* for 2 min.

### 2.4. Data Extraction and Analysis: Detecting IgA and IgG Responses

The slides were scanned with InnoScan 710 scanner (Innopsys, Carbonne, France) at 10 µm resolution, 5% gain and 5 mW power for the 635 nm laser and 10% gain and 10 mW power for the 532 nm laser to generate 16-bit TIFF images, and the raw data (.gpr file format) extracted with the MAPIX software Version 8.1.1 (Innopsys, Carbonne, France).

Data pre-processing was done using an in-house developed R script. For each spot, the neighborhood background relative fluorescent units (RFU) was subtracted from the foreground RFU to obtain a net RFU. A background threshold was applied whereby any foreground signals ≤ neighborhood background plus 2 standard deviations were zeroed. The mean-value, standard deviation and co-efficient of variation (CV) was obtained for triplicate spots of each protein. If the CV ≥20%, the CV of the two spots with the lowest CV is obtained. If the CV remains ≥20% for any probes, the sample is re-assayed. The reciprocal titer was calculated as described before and the cumulative titer calculated for the N protein by summing the RFU values for epitope 1, epitope 2, epitope 3, NTD and CTD; N protein epitopes and construct sequences has been published [10]. Cohort threshold values for the N protein constructs were obtained using the OptimalCutpoints package with an emphasis on maximizing specificity [11], whereas the threshold-values for the S protein was obtained using the average plus twice the standard deviation of the pre-pandemic samples. Samples with RFU-values greater than these N and S protein thresholds were considered IgA- and/or IgG-positive for the respective protein.

The longitudinal anti-S IgA and IgG response in participants were assessed based on whether they had a prior SARS-CoV-2 infection or not, and whether they received the placebo or the ChAdOx1 nCoV-19 vaccine.

### 2.5. Gene Synthesis and Cloning of the SARS-CoV-2 Ancestral, B.1.1.7 and B.1.351 Spike Protein Variants

The cDNA of SARS-CoV-2 S genes, positions 1-1273, of MN908947.3 (Wuhan-hu-1, China), B.1.1.7 (Kent/UK/204820464/2020) (Alpha) and B.1.351 (Port Elizabeth/South Africa/KRISP-K005325/2020) (Beta) were chemically synthesized (GeneArt, ThermoFisher Scientific). For the recombinant expression of correctly assembled intact S protein in vivo, R682S mutation was included with the synthesized cDNA sequences to prevent the furin mediated S1/S2 cleavage within the S protein.

The cDNA was cloned into a proprietary Escherichia coli/Spodoptera frugiperda transfer vector, pPRO8, such that the construct encoded the full-length N protein as an in-frame fusion to a C-terminal Biotin Carboxyl Carrier Protein (BCCP) fused to a c-Myc. pPRO8 is a derivative of pTriEx1.1 (Sigma, St Louis, MO, USA) and encodes the E. coli BCCP domain (amino acids 74–156 of the E. coli accB gene) downstream of a viral polyhedrin promoter and cloning sites; flanking this polh-BCCP expression cassette are the baculoviral 603 gene and the 1629 genes to enable subsequent homologous recombination of the construct into a replication-deficient baculoviral genome [12].

### 2.6. Derivatisation of ACE2 with Alexa Fluor 647 NHS Ester

The vial of 179 µL recombinant ACE2 (Raybiotech) was buffer exchanged thrice with 500 µL 0.1M sodium bicarbonate (Glentham Life Sciences) using the 0.5 mL 30 K Amicon Ultra centrifugal columns (MERCK) at 15,000× *g* for 10 min, with the final volume at 20 µL remaining in the spin column. Immediately before use, the Alexa fluor 647 NHS ester (ThermoFisher Scientific) was diluted to 10 mg/mL in DMSO (Glentham Life Sciences) and 2 µL was added to the recombinant ACE2 in the spin column. The spin columns were incubated for 1 h at room temperature with gentle shaking and protected from light. Thereafter, the protein was buffer-exchanged thrice with 500 µL phosphate buffered saline (PBS) (Gibco) using the 0.5 mL 30 K Amicon Ultra centrifugal columns (MERCK) at 15,000× *g* for 10 min, with the final volume at 20 µL remaining in the spin column. The protein concentration and protein-to-dye ratio was determined by nanodrop (NanoDrop One, Thermo Scientific), and the protein was diluted to 2 µM in PBS and stored in 15 µg aliquots at −20 °C.

### 2.7. Microarray Assays: Neutralization Assay of Serum Samples, and Detecting Recombinant Protein Spots with Anti-His Antibody

The S protein variant microarray slides were removed from −20 °C storage and washed twice for 5 min in wash buffer and twice for 5 min wash in PBS only, then dried by centrifugation at 1200× *g* for 2 min, and assembled in cleaned 24-plex gaskets. The serum samples were removed from −80 °C, thawed on ice then diluted 1:50 in ACE2 buffer (6 nM AF647-ACE2 in PBS, 0.2% Tween-20, 1% BSA and 1% milk powder) and incubated on the microarray for 1 h at room temperature shaking at 100 RPM. The slides were subsequently washed thrice for 5 min in wash buffer, and incubated with 5 µg/mL Cy3-anti-human IgG (in incubation buffer) for 30 min. Microarrays were rinsed thrice, and then twice for 5 min in wash buffer and twice for 5 min wash in PBS only, then dried by centrifugation at 1200× *g* for 2 min. Each recombinant protein probe on the microarray, which contains a his-tag, was detected by incubating a single microarray with 20 µg/mL Alexa fluor 647-anti-6xhis antibody (ThermoFisher) in incubation buffer for 30 min at room temperature, shaking at 100 RPM. Microarrays were rinsed thrice, and then washed twice for 5 min in Wash buffer and twice for 5 min in PBS only, then dried by centrifugation at 1200× *g* for 2 min.

### 2.8. Data Extraction and Analysis: Neutralization Assay for Serum Samples

The slides were scanned with the InnoScan 710 scanner (Innopsys, Carbonne, France) at 10 µm resolution, 50% gain and 10 mW power for the 635 nm laser and 5% gain and 10 mW power for the 532 nm laser to generate 16-bit TIFF images. The microarray was incubated with the Alexa fluor 647-anti-6xhis antibody was scanned at 10 µm resolution, 50% gain and 10 mW power with the 635 nm laser to generate 16-bit TIFF images. All raw data (.gpr file format) were extracted with the MAPIX software Version (Innopsys, Carbonne, France). Data pre-processing was done as described above. Thereafter, the amount of recombinant S protein of the ancestral, B.1.1.7 and B.1.351 protein loaded on the microarray chip was normalized using the relative intensities for each protein obtained from the anti-his assay.

The longitudinal change in the anti-S IgG levels was determined by the change in IgG for the RFU, whereas the ID50-values were assessed for ACE2 binding; the ID50-values were obtained by dividing 50% of ACE2 RFU binding in a buffer only control (i.e., no serum) by the ACE2 RFU binding for the serum sample multiplied by the serum dilution factor. The ID50-value is therefore inversely proportional to the amount of ACE2 bound to the S protein. The threshold for participants with an IgG response and ACE2 blockade was the same as determine for IgA and IgG responses (described above). The percentage of patients that have antibodies that block ACE2 binding was assessed for the S protein of the ancestral, B.1.1.7 and B.1.351 SARS-CoV-2 strains to assess vaccine efficacy against multiple variants.

## 3. Results

### 3.1. ChAdOx1 nCov-19 Vaccine Produces Anti-S IgG, but Not Anti-S IgA, in Vaccinated PLWH

Baseline anti-S IgA levels were detected throughout the time course for uninfected placebo receiving (D0 = 1448.0 RFU; D28 = 1585.7 RFU; D42 = 1396.1 RFU), uninfected vaccinated (D0 = 1415.6 RFU; D28 = 1420.6 RFU; D42 = 1401.0 RFU), infected placebo receiving (D0 = 1686.2 RFU; D28 = 1614.4 RFU; D42 = 1882.6 RFU), and infected vaccinated (D0 = 1545.0 RFU; D28 = 1675.0 RFU; D42 = 1554.0 RFU) PLWH (Figure 1). In contrast, a group of HIV-negative uninfected placebo-receiving (D0 = 1531.1 RFU; D28 = 1636.7 RFU; D42 = 1551.4 RFU) and vaccinated (D0 = 1446.3 RFU; D28 = 1477.6 RFU; D42 = 1330.8 RFU) participants produced baseline IgA, but infected placebo-receiving (D0 = 1928.8 RFU; D28 = 1795.2 RFU; D42 = 2237.7 RFU) and vaccinated (D0 = 1256.9 RFU; D28 = 2040.4 RFU; D42 = 1256.9 RFU) participants have increased IgA levels over the time course (Appendix A). Together these results indicate that PLWH might produce a short-lived or no anti-S IgA in response to infection, vaccination or a combination of the two.

The uninfected placebo-receiving group produced baseline anti-S IgG levels throughout time course (D0 = 3772.7 RFU; D28 = 3698.8 RFU; D42 = 3408.2 RFU), whereas the uninfected vaccinated group had increased IgG from D28 (4116.8RFU), which increased further at D42 (4543.4 RFU), indicating that the ChAdOx1 nCov-19 vaccine successfully induces IgG production after 1 dose, which further increases after 2 doses in PLWH. The infected placebo-receiving group had increased IgG levels at D28 (5159.9 RFU) which plateaued; interestingly, these IgG levels were higher than the IgG levels after 2 doses in uninfected vaccinated individuals. The infected vaccinated individuals produced the highest IgG levels after 1 vaccine dose at D28 (10667.2 RFU), which plateaued, suggesting that vaccination with a prior infection results in better protection than a previous infection or vaccination alone (Figure 1).

The uninfected participants did not produce an anti-N IgA or IgG response; however, all infected groups had decreasing antibody levels to the N protein (Appendix A). The percentage of participants with an anti-S IgA or IgG response was also assessed and revealed, much like the RFU values, that uninfected participants produce an IgG response post-vaccination, and that a prior infection results in more participants with an IgG response, and that a prior infection with vaccination results in 100% of patients producing IgG against the S protein (Appendix A).

### 3.2. ChAdOx1 nCoV-19 Vaccination with a Previous Infection Results in More Antibodies That Blocks ACE2 Binding than Vaccination Alone for the Ancestral, B.1.1.7 and B.1.351 S Protein Variants

Figure 2A represents the anti-S IgG response of participants towards the ancestral, B.1.1.7 and B.1.351 variants. Uninfected placebo-receiving participants did not produce an increased IgG response to the S proteins for the ancestral, B.1.1.7 or B.1.351 strains; however, uninfected vaccinated participants produced increased IgG to the S protein for the ancestral strain over the time course, and to a lesser extent towards the B.1.1.7 and B.1.351 strains. Infected placebo-receiving patients increased IgG towards the ancestral strain, and a lower response to the B.1.351 strain. Infected vaccinated participants produced high IgG levels in response to the ancestral S protein from D28, which plateaued. Interestingly, IgG was also produced against the B.1.1.7 and B.1.351 variants, but to a higher extent against the B.1.351 variant S protein, suggesting that the ChAdOx1-nCoV19 might result in better protection against the B.1.351 than the B.1.1.7 variant in PLWH. Figure 2B represents the ACE2 blockade of antibodies of participants towards the S protein of the ancestral, B.1.1.7 and B.1.351 SARS-CoV-2 variants. Uninfected placebo-receiving participants did not produce antibodies that block ACE2 binding, however, uninfected vaccinated participants produced a marginal increase in the levels antibodies that block ACE2 binding. Similarly, infected placebo-receiving participants did not produce antibodies that block ACE2 binding; however, infected vaccinated participants produced antibodies that block ACE2 binding to the S protein for all strains, but highest binding to the ancestral strain, then the B.1.351 and then the B.1.1.7. These results indicate that while infection results in greater anti-S IgG production than vaccination, the vaccination induces the production of more neutralizing antibodies.

The percentage of participants that both produced IgG and blocked ACE2 binding was assessed to determine the percentage of participants that produce neutralizing antibodies (Figure 3). The percentage of uninfected placebo-receiving patients remained low and largely unchanged. The percentage of uninfected vaccinated participants with IgG that block ACE2 binding increased over the time course with the greatest increase for the ancestral S protein, suggesting that the vaccine may offer greatest protection against the ancestral strain, but blockade against the B.1.1.7 was seen from D28 and B.1.351 from D42, suggesting that ChAdOx1-nCoV19 vaccine may offer protection against these, and perhaps other strains as well. The percentage of infected placebo-receiving patients that had anti-S IgG and ACE2 blockade was relatively high for all strains. The percentage of participants with anti-S IgG and ACE2 blockade at D0 are similar for infected placebo-receiving and vaccinated participants; however, the percentage increase markedly for vaccinated participants at D28 (ancestral = 93.3%; B.1.1.7 = 93.3%; B.1.351 = 80%) compared to placebo-receiving patients (ancestral = 47.1%; B.1.1.7 = 47.1%; B.1.351 = 17.6%) and reaches 100% for ancestral, but decreases to 86.7% for B.1.1.7 and 80% for B.1.351, after the second dose. These results support the hypothesis that a 1 dose of the ChAdOx1 nCov-2 vaccine with a previous infection, results in greater protection than infection or ChAdOx1 nCov-19 vaccination alone.

## 4. Discussion

Reports on the effects of SARS-CoV-2 vaccines in PLWH are increasing [13], but still scarce. In this study we present the quantitative IgA and IgG responses, as well as ACE2 binding blockade, against S protein variants in Black PLWH in response to the ChAdOx1 n-Cov19 vaccine. Here, we report for the first time that, in contrast to their HIV-negative counterparts, PLWH did not produce a detectable anti-S IgA response with an infection or ChAdOx1 nCoV-19 vaccination. Previous reports indicate that while IgG production and neutralization becomes dominant later, the IgA response reportedly persists 28 days post-symptom onset [14], and ChAdOx1 nCov19 vaccination reportedly induces anti-S IgA production in study volunteers, which peaks at 28 days post vaccination [15]. The anti-S IgA response for HIV-negative participants indicate that an anti-S IgA response is detectable by our assays, but were not detected for both infected and vaccinated PLWH. A previous study reported that serum anti-RBD IgA response precedes the IgG response to SARS-CoV-2 infection, and is reportedly more potent in their neutralization compared to IgG in early stage SARS-CoV-2, i.e., the first week after symptom onset [14]. This means that PLWH might not have the early stage IgA-induced neutralization—and therefore likely protection afforded thereby—of their HIV-negative counterparts. Whether the absence of measurable anti-S IgA response in blood following vaccination or natural infection in PLWH observed here reflects a fundamental difference in B-cell responses in PLWH due to perturbation of CD4+ T-cell responses, or whether the anti-S IgA response is simply shorter-lived in PLWH than in HIV-negative participants remains to be determined. Follow-on studies will also be needed to explore whether an anti-S IgA response is detectable at the site of disease, i.e., the lungs, following natural infection, as well as whether absence of an anti-S IgA response in blood or mucosal samples correlates with increased susceptibility to SARS-CoV-2 infection in PLWH.

In contrast, an anti-S IgG response was detected for both infected and vaccinated PLWH; however a previous infection results in greater IgG production than vaccination alone, and that a vaccination with a previous infection results in greater IgG levels, indicating that ChAdOx1 nCov-19 vaccinated PLWH still have serum IgG protection. These results are consistent with a previous ChAdOx1 nCoV-19 study, which showed that anti-S IgG was detected for PLWH and HIV-negative participants; however, that study did not stratify participants based on a previous infection [16]. A previous study revealed that the concentration of neutralizing antibodies predicts protection against SARS-CoV-2 infection [17]. Thus, assessing the presence and change in neutralizing antibodies in vaccinated individuals may predict protective outcomes, which is especially important for at-risk groups, including PLWH. Our results indicate that ChAdOx1-nCoV2 vaccination results in anti-S IgG production against the ancestral, but to a much lower extent against the B.1.1.7 and B.1.351 strains. Interestingly, the vaccine results in greater anti-S IgG production against the B.1.351 than the B.1.1.7 strain, suggesting that the B.1.1.7, and perhaps other strains with similar mutations, may escape ChAdOx1 nCoV-19 vaccine-induced protection more efficiently in PLWH. Previous reports using pseudovirus and live-virus neutralisation assays on larger number of patients in this cohort also revealed that the ChAdOx1 nCoV-19 vaccine has reduced or abrogated neutralization against the B.1.351 strain [18], which we also observed with our ACE2 blockade assay. Thus, the on-array ACE2 binding assay used here represents a technically simple, high throughput way to rapidly quantify the direct impact of spike variants on antibody blockade of spike-ACE2 binding, as a surrogate neutralization assay. Our results further revealed that while infection and vaccination resulted in anti-S (ancestral) IgG, only vaccinated participants produced increasing antibody levels that block ACE2 binding. These findings indicate that the ChAdOx1 nCoV-19 vaccine resulted in the production of neutralizing antibodies, but that ACE2 blockade was marginal in vaccinated PLWH. Importantly, the results indicate that vaccination with a previous infection results in the highest IgG levels which is also reflected in the extent of ACE2 ID50-values for the same patients. Together, these results indicate that one dose of the ChAdOx1 nCoV-19 vaccine with a previous infection offers greatest protection in PLWH.

In conclusion, this study has revealed that PLWH produce an anti-S IgG, but not an anti-S IgA, response to SARS-CoV-2 infection, ChAdOx1 nCoV-19 vaccination, or both. Moreover, individuals with a previous infection and one ChAdOx1 nCoV-19 vaccine dose produced more anti-S IgG, and blocked ACE binding to the S protein more efficiently, than individuals who were only infected or only vaccinated.

## Figures and Tables

**Figure 1 viruses-15-00448-f001:**
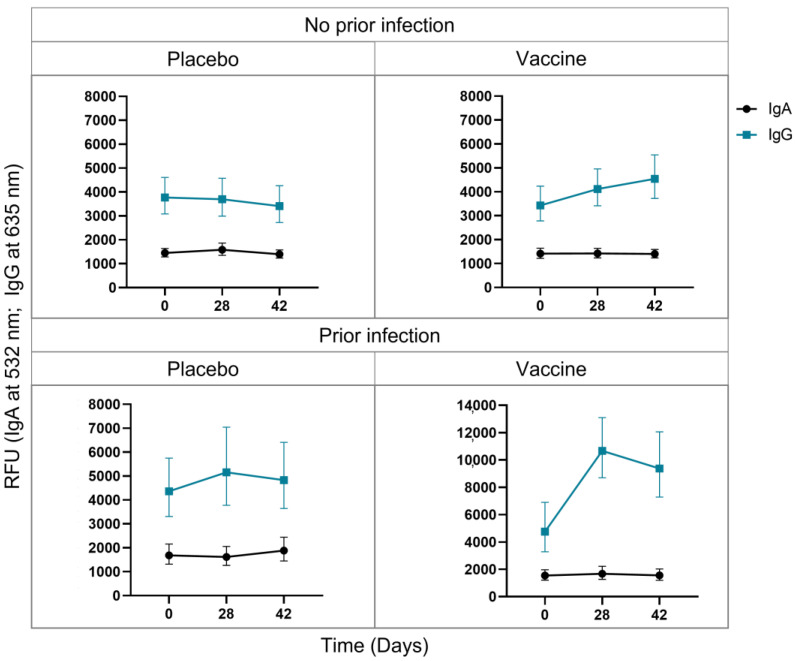
Longitudinal (Day 0, Day 28 and Day 42) change in IgA and IgG against the SARS-CoV-2 ancestral spike (S) protein in response to the ChAdOx1 nCoV-19 vaccinated in Black people living with HIV (PLWH). Participants were vaccinated at Day 0 and Day 28, and serum collected before vaccine administration at Day 0, Day 28 ad Day 42, and assessed based on division into a placebo or vaccine-receiving groups, with or without a prior infection. The average relative fluorescence unit (RFU) is represented and the error bars indicate the minimum and maximum RFU values.

**Figure 2 viruses-15-00448-f002:**
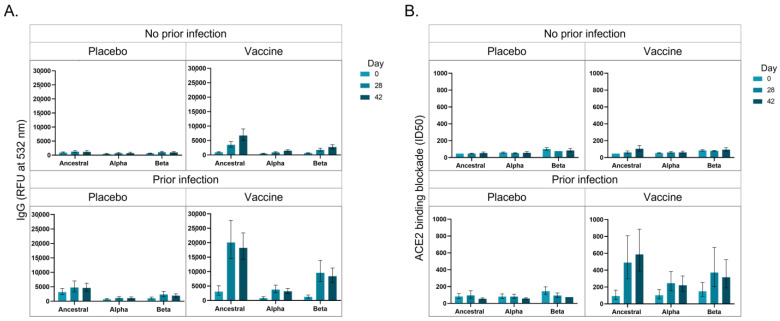
The longitudinal (Day 0, Day 28 and Day 42) IgG production (532 nm) (**A**) and ACE2 binding blockade (ID50) (**B**) against SARS-CoV-2 spike (S) protein variants (ancestral, Alpha/B.1.1.7 and Beta/B1.351) in Black people living with human immunodeficiency virus (HIV) (PLWH) is depicted. Participants were vaccinated at Day 0 and Day 28, and serum collected before vaccine administration at Day 0 and Day 28, as well as Day 42. Participants were assessed based on whether they were vaccinated or received the placebo, with or without a previous SARS-CoV-2 infection. The average relative fluorescence units (RFU)- or ID50-values are represented and the error bars indicate the minimum and maximum RFU- or ID50-values.

**Figure 3 viruses-15-00448-f003:**
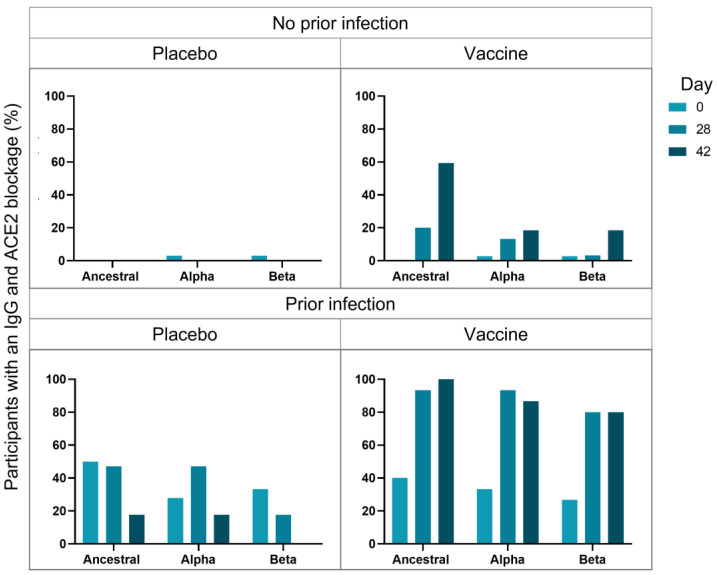
The longitudinal (Day 0, Day 28 and Day 42) change in the percentage of participants with both an IgG response and who have antibodies that blocked ACE2 binding to the ancestral, Alpha/B.1.1.7 and Beta/B.1.351 SARS-CoV-2 spike (S) protein variants are depicted. Participants were vaccinated at Day 0 and Day 28, and serum collected before vaccine administration at Day 0, Day 28 and Day 42, and assessed based on division into a placebo or vaccine-receiving groups, with or without a prior infection.

## Data Availability

Data are contained within the article or Appendix A.

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
