# Peer review of "Longitudinal IgA and IgG Response, and ACE2 Binding Blockade, to Full-Length SARS-CoV-2 Spike Protein Variants in a Population of Black PLWH Vaccinated with ChAdOx1 nCoV-19"

_viruses, 2023, doi:10.3390/v15020448_

Round 1

Reviewer 1 Report

Dear all,

your manuscript is well written , an interesting subject but I consider it can be improved in English language a little

Reviewer 2 Report

A more extended "discussions" session would be beneficial as well as a paragraph of conclusions.

Reviewer 3 Report

The authors have presented interesting results in a study looking at longitidinal IgA and IgG responses and temporal patterns in face of encountering full-length SARS-CoV-2 spike protein variantin in a population on PLWH (immunocompromised cases) already vaccinated with ChAdOx1 nCoV-19. 

While the study is designed well and is on an important and timely topic, the authors need to improve the paper by addresing the following issues:

1. The figures must be in colosr and all in formation should be in the plots (time scales, ....). Current figures are not informative.

2. Authors need to highlight the novel findings they have reached in more details.
